# Improving extractive document summarization with sentence centrality

**Shuai Gong**[1], **Zhenfang Zhu**[1]*, **Jiangtao Qi**[1], **Chunling Tong**[1], **Qiang Lu**[2], **Wenqing Wu**[1]

**1** School of Information Science and Electrical Engineering, Shandong Jiaotong University, Jinan, Shandong Province, China, **2** School of Information Science and Electrical Engineering, Northwest University, Xian, Shanxi Province, China

* zhuzf_sdjtu@126.com

**Data Availability Statement:** All relevant data are within the article and its Supporting information files.

## Abstract

Extractive document summarization (EDS) is usually seen as a sequence labeling task, which extracts sentences from a document one by one to form a summary. However, extracting sentences separately ignores the relationship between the sentences and documents. One solution is to use sentence position information to enhance sentence representation, but this will cause the sentence-leading bias problem, especially in news datasets. In this paper, we propose a novel sentence centrality for the EDS task to address these two problems. The sentence centrality is based on directed graphs, while reflecting the sentence-document relationship, it also reflects the sentence position information in the document. We implicitly strengthen the relevance of sentences and documents by using sentence centrality to enhance sentence representation. Notably, we replaced the sentence position information with sentence centrality to reduce sentence-leading bias without causing model performance degradation. Experiments on the CNN/Daily Mail dataset showed that EDS models with sentence centrality significantly improved compared with baseline models.

## Introduction

Automatic document summarization aims to produce a concise summary of a document while preserving its crucial information. Existing summarization methods can be divided into two categories: abstractive and extractive methods. Abstractive methods generate a summary word by word from scratch, and can introduce new words that do not appear in the document [1]. Extractive methods, on the other hand, form a summary by selecting text fragments from the original document. Compared with abstractive methods, extractive methods are inclined to generate semantically and grammatically correct sentences [2, 3].

In recent years, extractive document summarization (EDS) based on neural networks has achieved great success [4–6]. However, it faces a challenge in modeling the sentence-document hierarchical structure. Previous approaches to solve this problem can be divided into two categories: (1) constructing hierarchical structures to represent documents and sentences

**Funding:** This study was funded by a grant from National Social Science Fund of China (19BYY076) to ZZ.

separately; (2) using certain sentence-document information to enhance the representation of sentences.

There is much excellent work based on the first approach. For example, Zhang et al. [7] proposed a hierarchical transformer [8] called HIBERT to strengthen the relationship between sentences and documents. Xu et al. [9] applied the self-attention scores in the sentence-level Transformer to measure the importance of sentences. Jia et al. [10] employed the hierarchical attention mechanism to establish intersentence relations. Although the hierarchical models effectively capture the sentence-document relationship, complex model architectures and huge computing power requirements limit their actual use scenarios. Another approach usually uses the position information of the sentence in the document to enhance the sentence representation. This method is simple and effective but will cause the sentence-leading bias problem, which means that the extractive summarizer tends to select the leading sentences in the document. Sentence-leading bias will cause the model to rely excessively on sentence position information rather than semantic information when selecting sentences [11, 12]. In this paper, we replace the sentence position information with the sentence centrality information.

The sentence centrality is usually based on undirected graphs and is widely used in unsupervised extractive summarization tasks to identify salient sentences in a document [13, 14]. In the task, a document is represented as a graph, in which each node is a sentence, weights of the edges are measured by sentence similarities. The centrality of a sentence can be measured by simply computing its node's degree. This method can be described in Fig 1(a). The number in the node represents the position of the sentence in the document, the node size indicates the sentence centrality score. The centrality of the Third sentence is related to all other sentences. Although the sentence centrality based on undirected graphs can reflect the relationship between the sentence and the document, it does not include the sentence position information, which has been shown to have an essential role in the EDS task [11]. Zheng and Lapata [14] construct directed graphs to compute the sentence centrality. Their work shows that for a sentence, the similarity with the previous content will damage its centrality. Inspired by their work, we calculate the centrality score of a sentence based only on the similarity between the sentence and its following content. Our approach to calculating the sentence centrality is presented in Fig 1(b).

Previous work considered sentence centrality as a signal to measure the importance of sentences [13–15]. Different from their work, we regard the sentence centrality as a unique property of the sentence, just like sentence position information in the document. Therefore, we use the sentence centrality to enhance sentence representation.

Following our intuition mentioned before, we can learn that the sentence centrality is no longer restricted in unsupervised extractive methods. We develop two methods to apply

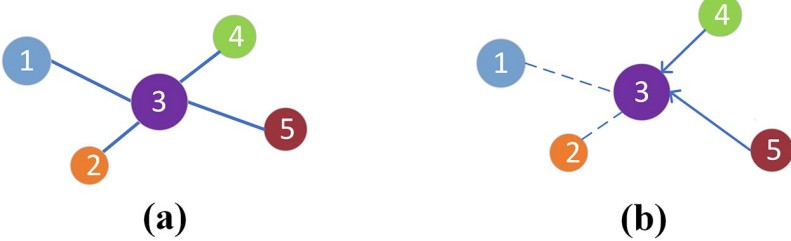

**(a)** **(b)**

**Fig 1. The sentence centrality based on the undirected graph (a) and the directed graph (b).** Figure (a) shows the conventional method for computing sentence centrality, and (b) is our method. In our method, we calculate the centrality of a sentence not by calculating the similarity of the sentence to all sentences in the document but only for some specific sentences.

sentence centrality to enhance sentences representation: (1) embedding the sentence centrality directly into the sentence representation output by the encoder; (2) updating the sentence representation indirectly via Graph Attention Network (GAT) [16]. We build two models to implement these two ideas. We firstly construct our sentence centrality-enhanced EDS model based on BERT. The model contains a BERT encoder and a summarization layer classifier to select sentences. Notably, in the summarization layer, we only use a simple linear classifier and do not use other methods such as Inter-sentence Transformer [17], RNN to enhance the model. Then, we construct the sentence centrality on the heterogeneous graphical extractive summarization model [18]. In the heterogeneous graph neural network, the sentence representation is varied by the attention mechanism. We extend the edge features with sentence centrality and use it to modify the GAT. The experimental results show that the performance of both models is significantly improved with the sentence centrality. Finally, we analyzed the position distribution of the sentence centrality in the document and explained why the sentence centrality information is practical.

The contributions of our work are as follows:

1. We propose a novel sentence centrality for EDS task and two approaches to use sentence centrality to enhance sentence representation. With the help of the sentence centrality, the relationship between sentences and documents is implicitly strengthened, thus improving the performance of the extractive summarization.

2. We propose to replace sentence position information with sentence centrality, which can reduce the sentence-leading bias in the news dataset caused by position information.

The remainder of this article is arranged as follows. We introduce some related topics on the EDS in the section Related Work. In the Method section, we define the EDS and then introduce our sentence centrality-enhanced extractive summarization models. We present the training details, parameter settings and experimental results in the Experiment section. In the Discussion section, we discuss why sentence centrality works. Finally, we conclude our paper in the Conclusion section.

## Related work

To make the paper self-contained, we will introduce some related topics on the EDS and the sentence centrality-based summarization methods.

### Extractive document summarization

The EDS task aims to extract sentences from the original document to form a summary. The task first encodes the sentences with the help of an encoder to obtain a sentence vector. The sentence vector is then passed through a classification layer to determine whether it should be included in the summary. Nallapati et al [2]; Zhou et al. [19] choose recurrent neural networks (RNN) for sentence encoding, while Wang et al. [20] use Transformer [8]. BERT and other pre-trained language models [21] also perform well in the EDS task. Besides, graph neural networks also have received extensive attention. Yasunaga et al. [22] apply the graph neural network for multi-document summarization. Wang et al. [18] propose to use the heterogeneous graph for the EDS task.

Although these methods are effective, they mostly rely on sentence position information to enhance sentence representation. We introduce sentence centrality information in the model and remove sentence position information, which improves model performance and does not cause sentence-leading bias.

## Sentence centrality-based summarization methods

The sentence centrality is often used to measure the importance of a sentence in unsupervised EDS tasks. In the task, a document is represented as a graph, with nodes representing sentences and edges connecting sentences weighted according to their similarity. TextRank [13] calculates similarity by analyzing the cooccurrence of words, LexRank [15] incorporates TF-IDF values into the weights of the edges, Zheng and Lapata [14] use BERT to measure sentence similarities.

There are three key differences in our sentence centrality compared to the previous methods. (1) We calculate the centrality of a sentence by counting only the similarity between that sentence and the content that follows it, not the similarity of all other content. (2) The centrality of a sentence is considered a unique property of the associated sentence and document rather than just as a measure of the importance of the sentence. Therefore, we use sentence centrality to strengthen the sentence representation. (3) We applied sentence centrality into the supervised EDS.

## Sentence embeddings for extractive document summarization

An essential step in the extractive document summarization task is to obtain sentence embeddings. Traditional sentence embedding methods are based on weighting and averaging words vectors to construct sentences' vectors. Kedzie et al. [23] averaged word embeddings of a sentence to get the sentence embedding. This method regards each word as having the same effect on the sentence and does not consider the specificity of particular words. Nallapati et al. [2] apply RNN to compute the hidden state representation at each word position sequentially, based on the current word embedding and previous hidden states, then use the average-pooled, concatenated hidden states as sentence embeddings. Compared with a simple average of words embeddings representation, Nallapati et al. [2] consider the order of words.

Traditional sentence embedding methods are simple and effective. However, extractive document summarization is a document-level task, and the relationship between sentences and documents needs to be considered when obtaining sentence embeddings. Most works [3, 10, 20, 21] strengthen sentence embeddings using the position information of sentences in the document.

Different from their work, we use sentence center information to enhance sentence representations. Compared to using sentence position information, our methods are able to achieve performance improvements while reducing the sentence lead bias problem.

## Method

We define the problem of EDS as follows. Given a single document $d$ that contains $n$ sentences, $d = \{s_1, s_2, \ldots, s_n\}$, where $s_i = \{w_{i1}, w_{i2}, \ldots, w_{im}\}$ is the $i$-th sentence in the document and $w_{ij}$ is the $j$-th word in the $i$-th sentence. EDS can be seen as a sequence labeling task [5], which means that every sentence in the document is assigned a label $y_i \in \{0, 1\}$ to suggest whether the sentence should be included in the summaries.

We introduce sentence centrality into the EDS task. The sentence centrality is used to enhance sentence representation in two ways. One is embedded directly into the sentence representation output by the encoder, and the other is to update the sentence representation indirectly via Graph Attention Network (GAT) [16]. In this section, we will first introduce the computation of the sentence centrality and then present our sentence centrality-enhanced EDS models.

## Calculation of sentence centrality

The first step of calculating the sentence centrality is to obtain the representations of sentences. We use BERT [24] as the sentence encoder. BERT is a recently proposed highly effective model that is based on deep bidirectional Transformers and has achieved state-of-the-art performance in many NLP tasks. BERT is fine-tune followed by Gao et al. [25] with contrastive learning:

$$l_i = log \frac{e^{sim(h_i, h_i^+)/t}}{\sum_{j=1}^{j=n} e^{sim(h_i, h_j)/t}},$$

(1)

where $h_i$ and $h_i^+$ are different vector representations of the same sentence, $sim(h_i, h_i^+)$ is the cosine similarity $\frac{h_i^T h_i^+}{\|h_i\| \cdot \|h_i^+\|}$, $t$ is a temperature hyperparameter. We feed the same sentences to the encoder twice by applying random dropout to get $h_i$ and $h_i^+$. After we get representations $\{sv_1, sv_2, \ldots, sv_n\}$ for sentence $\{s_1, s_2, \ldots, s_n\}$ in the document $d$, we calculate centrality of sentence $s_i$ by following these steps. Firstly, we employ paired dot product to compute the similarity matrix $E_i$ for sentence $s_i$:

$$E_{ij} = (sv_i)^T sv_j (i \neq n, j > i).$$

(2)

Then, we calculate the centrality of the sentence $s_i$ (Shorthand for $SC_i$ by averaging the elements included in $E_i$:

$$SC_i = \frac{1}{n-i-1} \sum_{j=i+1}^{j=n} e_{ij}.$$

(3)

Through the Eqs (2) and (3), we can obtain the sentence centrality $\{SC_1, SC_2, \ldots, SC_{(n-1)}\}$ for the sentence $\{s_1, s_2, \ldots, s_{(n-1)}\}$. Note that the centrality of the last sentence in the document is not calculated, we average other $n-1$ sentences' centrality to get $SC_n$. That seems counter-intuitive since the last sentence should intuitively summarize the articles and have a high sentence centrality score. In fact, information in the last sentence is not as much as we intuitively think because of the particularity of the news dataset [20]. So far, we have obtained the centrality of all sentences in one document: $SC_d = \{SC_1, SC_2, \ldots, SC_n\}$. We normalize $SC_d$ by the following way:

$$\widetilde{SC_i} = \frac{SC_i - Min(SC_d)}{Max(SC_d) - Min(SC_d)}.$$

(4)

The centrality of all sentences in a document is ultimately expressed as:

$$SC_d = \{\widetilde{SC_1}, \widetilde{SC_2}, \ldots, \widetilde{SC_n}\}.$$

(5)

## Sentence Centrality-enhanced EDS models

We build two models to implement the previously mentioned methods for enhancing sentence representations separately. The first model is based on BERT to realize viewpoint one: directly embedding sentence centrality into the sentence representation, and the second model is based on the heterogeneous graph neural network EDS model (HSG) of Wang et al. [18] to realize the viewpoint two: modifying the attention mechanism through sentence centrality and then indirectly enhancing the sentence representation through the attention. The method (2) could demonstrate the idea that the sentence centrality is a special property of sentences, because the sentence representation will update according to its centrality.

**Sentence centrality-enhanced EDS model based on BERT.** We firstly build our sentence centrality-enhanced EDS models based on BERT. The overview of this model is presented in Fig 2.

The model contains a BERT encoder and a summarization layer classifier. BERT is applied to obtain a contextual representation of each word for each sentence in the input document:

$$[u_{11}, u_{12}, ..., u_{nm}] = BERT([w_{11}, w_{12}, ..., w_{nm}]). \tag{6}$$

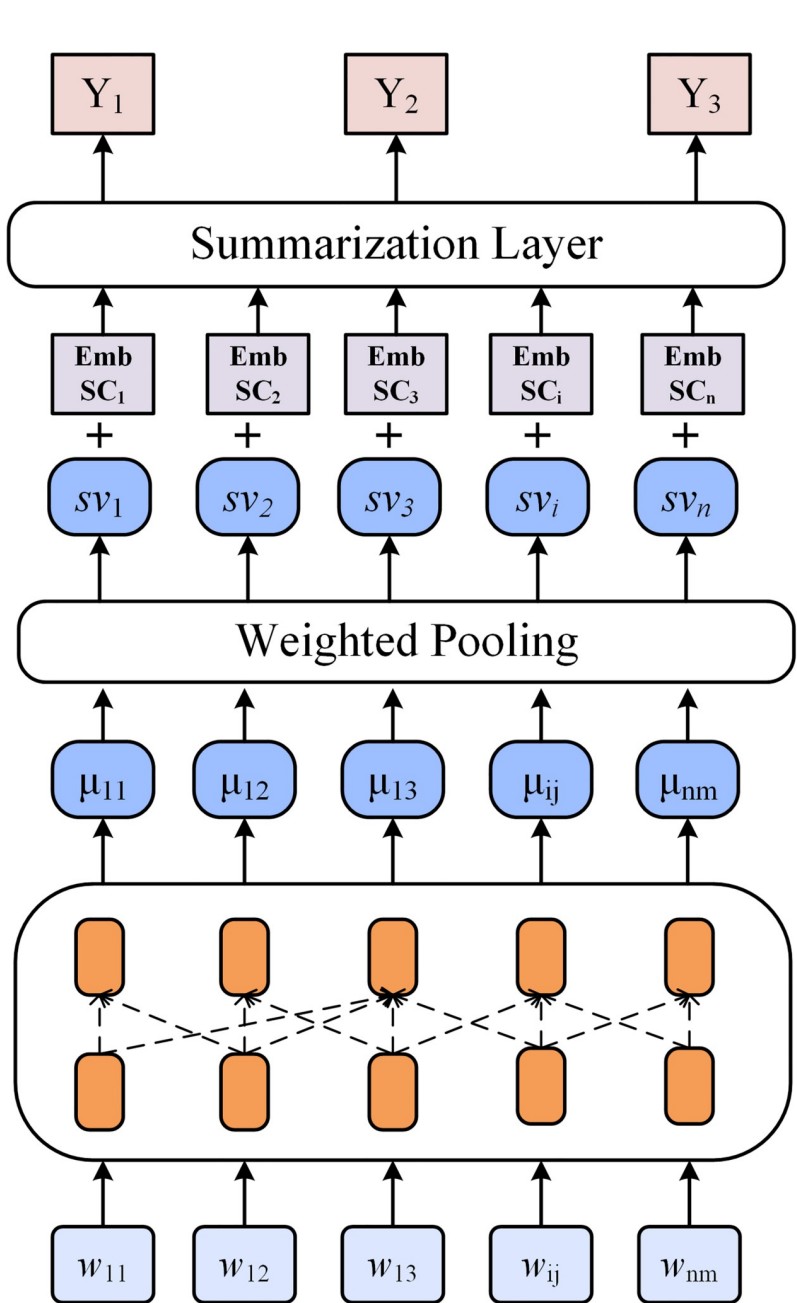

**Fig 2. Sentence centrality-enhanced EDS model based on BERT.** *EmbSC_i* is the centrality embedding of sentence $s_i$, which is directly embedded in the sentence representation generated by BERT. The sentence position embedding is replaced by sentence centrality embedding.

where the $u_{ij}$ is the contextual representation for $w_{ij}$. The sentence's representation is obtained by weighted pooling:

$$a_{ij} = \mathbf{W}(u_{ij})^T. \tag{7}$$

$$sv_i = \frac{1}{n}\sum_{j=1}^{n} a_{ij}u_{ij}. \tag{8}$$

In this way, we obtain the vector representation for each sentence in the document. Then, we obtain the sentence centrality embedding ($EmbSC_i$) by mapping the normalized scalar sentence centrality to the multi-dimensional embedding space:

$$EmbSC_i = \mathbf{W_{sc}}\widetilde{SC}_i. \tag{9}$$

where $\mathbf{W_{sc}}$ is a weight matrix with the weights set to 1. $EmbSC_i$ is the centrality embedding of sentence $s_i$, which has the same dimension as the sentence embedding. The final vector representation of sentence $s_i$ in the document is represented as:

$$h_i = sv_i + EmbSC_i, \tag{10}$$

where $sv_i$ is the vector representation of the sentence $s_i$ output by BERT.

In the summarization layer, we only use a simple classifier and do not use other methods such as Inter-sentence Transformer [17], RNN [2] to enhance the model. The simple classifier only adds a linear layer on the final sentence vector representation and use a sigmoid function to get the predicted score:

$$\hat{Y} = \sigma(\mathbf{W_0} + \mathbf{b}). \tag{11}$$

where $\sigma$ is the sigmoid function, $\mathbf{W_0}$ is trainable weights matrix, $\mathbf{b}$ is a bias matrix. The loss of the model is the binary classification entropy of prediction $\hat{Y}$ against gold label $Y_i$.

**Sentence centrality-enhanced EDS model based on HSG.** Heterogeneous summarization graph (HSG) [18] is an extractive summarization model based on heterogeneous graph neural networks, which achieves the optimal performance in the architecture without pre-trained contextualized encoders. The model contains two kinds of nodes: word nodes and sentence nodes. The TF-IDF value of the word links the word nodes and the sentence node containing the words. We build sentence centrality on this model for two reasons: (1) it can be verified that sentence centrality is equally valid in the architecture without pre-trained contextualized encoder; (2) it serves the purpose of indirectly enhancing the sentence representation by modifying the attention mechanism.

Our modified HSG model is presented in Fig 3. The word embedding is obtained by a word encoder. Here, we use a 300-dimensional GloVe [26] embedding for each word in the sentence. We first use Convolutional Neural Networks (CNN) [27] with different kernel sizes to capture local n-gram features for each sentence, and then obtain sentence-level features using Bidirectional Long and Short-Term Memory (BiLSTM) [28]. We use graph attention networks (GAT) [16] to update the representations of our semantic nodes. The GAT layer is modified by infusing the scalar edge weights $e_{ij}$, which are mapped to the multi-dimensional embedding space. The weights of the edge $e_{ij}$ are the sum of the sentence centrality and the TF-IDF value of the words, because the types of nodes connected by the edge are different. The modified

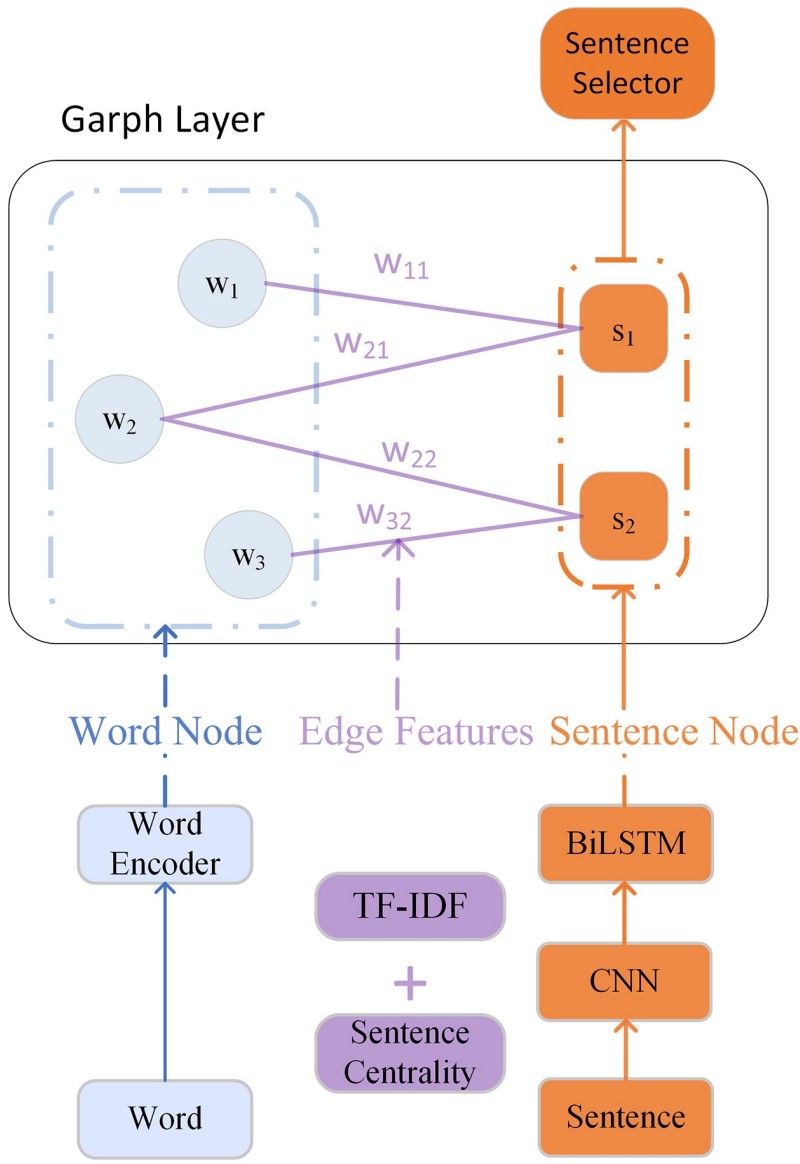

**Fig 3. Sentence centrality-enhanced EDS model based on HSG.**

GAT layer is designed as follows:

$$e_{ij} = TFIDF_i + \widetilde{SC_j}. \tag{12}$$

$$z_{ij} = LeakyReLU(\mathbf{W_a}[\mathbf{W_q}h_i; \mathbf{W_k}h_j; e_{ij}]). \tag{13}$$

$$\alpha_{ij} = \frac{exp(z_{ij})}{\sum_{l \in N_i} z_{il}}. \tag{14}$$

$$u_i = sigmoid\left(\sum_{j \in N_i} \alpha_{ij}\mathbf{W_v}h_j\right). \tag{15}$$

where $h_i$ is the hidden state of input node, $\alpha_{ij}$ refers to the weight of attention between $h_i$ and $h_j$. The residual connection is used to avoid gradient vanishing after several iterations. The final sentence representation is:

$$h_i^{'} = h_i + u_i. \tag{16}$$

Given a constructed graph $G$ with word features $X_w$ and sentence node features $X_s$, the sentence nodes are updated with their neighbor word nodes via the above GAT and feed-forward (FFN) layer:

$$U_s^1 = GAT(H_s^0, H_w^0, H_w^0). \tag{17}$$

$$H_s^1 = FFN(U_s^1 + H_s^0). \tag{18}$$

where $H_w^0 = X_w$, $H_s^0 = X_s$, $GAT(H_s^0, H_w^0, H_w^0)$ denotes that $H_s^0$ is used as the attention query and $H_w^0$ is used as the key and value. The updated sentence representations are then feed into the sentence selector moudle. In the sentence selector moudle, we do node classification for sentences and ues the cross-entropy loss as the training objective for the whole system.

## Experiment

### Dataset

We conduct our experiment on the CNN/Daily [29] Mail, Xsum [30] datasets.

CNN/Daily Mail is a well-known news dataset for single document extractive summarization, which is split into three parts by Hermann et al. [24] for training, validation, and testing. The splits contain 90,266/1,220/1,093 CNN documents and 196,961/12,148/10,397 Daily Mail documents. We process the dataset by the Stanford CoreNLP toolkit [31] following methods in See et al. [32].

XSum is a one-sentence summary dataset to answer the question "What is the article about?". We conduct experiments on this dataset to study whether sentence centrality-enhanced EDS models are still effective when dealing with dataset with short summaries.

We only use the XSum dataset for ablation experiments, as the extraction results on this dataset are few and are insufficient to support our model performance comparison.

### Implementation details

We limit the sentence length to 50 words calculating sentence centrality. Both models are trained on a single GPU (GeForce RTX 3080).

**Sentence centrality-enhanced EDS model based on BERT.** The model is implemented by the 'bert-base-uncased' version of BERT, which can be obtained in https://github.com/huggingface/pytorch-pretrained-BERT. The model is trained for 40000 steps. The best result on the validation set occurs at step 37000. Adam algorithm is applied for optimizing the loss function. Learning rate schedule is following Vaswani et al. [33] with warming-up on first 10,000 steps:

$$lr = 2e^{-3} \cdot min(step^{-5}, step \cdot warmup^{-0.5}) \tag{19}$$

We score the sentences and then select the top-3 sentences with the highest scores as the summaries.

**Sentence centrality-enhanced EDS model based on HSG.** The word nodes are initialized with $d_{emb} = 300$ while sentence nodes are $d_s = 128$. The dimension of edge feature $e_{ij} = 50$.

Each GAT layer is eight heads with a hidden state $d_h$ = 64. We train the model with 32 batch sizes for 20 epochs and use the Adam algorithm [34] to optimize the loss function with the learning rate $5e^{-4}$. In the decoding stage, we choose the three sentences with the highest scores as document summaries.

## Trigram blocking

In the prediction phase of both two models, we use Trigram Blocking [35] for decoding, a simple and practical approach to reducing redundancy. In the stage of selecting sentences to form a summary, it skips sentences that have triple overlap with previously selected sentences. Surprisingly, this simple method of removing duplication brings a remarkable performance improvement.

## Baselines and comparisons

We compare our models with the following solid baselines for text summarization:

*LEAD-3*: The method takes the first three sentences of the document as a summary.

*HSG* [18]: An extractive method based on the heterogeneous graph neural network. This method constructs the document as a heterogeneous graph. The graph contains two different types of nodes: sentence nodes and word nodes. Information can be passed between the nodes.

*JECS* [36]: A hybrid method. The method firstly selects sentences and then compresses each sentence by removing unnecessary words.

*LSTMPN* [37]: An extractive model based on LSTM and pointer network.

*LongformerExt* [38]: An extractive model based on Long Transformer. This method enables the complete input of sentences and documents to the encoder.

*BERTSUMEXT* [17]: A method based on the pretrained model BERT. The model encodes sentences by BERT and uses Inter-sentence Transformer to capture the document-level information further.

*PNBERT* [39]: An extractive model based on BERT and pointer network.

*BERTRL* [39]: The method encodes sentences by BERT and uses reinforcement learning to solve the problem of inconsistency between training and evaluation objectives.

*HIBERTM* [7]: An extractive model based on BERT. The model proposed a hierarchical transformer to strengthen the relationship between sentences and documents.

## Results

We test our model on the CNN/Daily Mail. ROUGE [40] scores measure the summarization quality. The definition of ROUGE scores is presented in S1 Appendix. The results of our BERT-based EDS model are presented in Table 1. Experimental results show a slight performance improvement of our sentence centrality-enhanced EDS model compared to BERT-SUM. The model BERTSUM uses Inter-sentence Transformer to strengthen the sentences-document relationship while we only use the sentence centrality, which could show that sentence centrality is effective in strengthening the relationship between sentences and documents.

**Table 1. The results of sentence centrality-enhanced EDS model based on BERT.**

| Model | ROUGE-1 | ROUGE-2 | ROUGE-L |
|---|---|---|---|
| LEAD-3 | 40.34 | 17.70 | 36.57 |
| PNBERT | 42.69 | 19.60 | 38.85 |
| BERTRL | 42.76 | 19.87 | 39.11 |
| HIBERTM | 42.37 | 19.95 | 38.83 |
| Longformer-Ext | 43.00 | 20.20 | 39.30 |
| BERTSUMEXT | 43.25 | 20.24 | 39.63 |
| SCBERT | **43.32** | **20.3** | **39.72** |

ROUGE scores measure the summarization quality. ROUGE-1, ROUGE-2, ROUGE-L are used for reporting the unigram, bigram, and longest common subsequence overlap with reference summaries. The first part presents the LEAD-3 baseline model. The second block shows the results of sentence-level extractors for comparison. SCBERT is our sentence centrality-enhanced EDS model based on BERT.

The experimental results of our model based on the heterogeneous graph neural network are shown in Table 2. The results show that the experimental performance on ROUGE-1, ROUGE-2, ROUGE-L outperforms all the models without pre-trained encoders.

## Ablation study

We performed ablation experiments to discuss the effects of the sentence centrality and sentence position on model performance. Experiments are conducted on CNN/Daily Mail and XSum. The models are presented as follows.

*SCES*: Extractive summarizer with the sentence centrality. We build our extractive summarization model based on BERT. We discard the position embedding of sentences, and the sentence centrality embedding is applied instead.

*SCPES*: Extractive summarizer with the sentence and position information. In this model, we do not discard the sentence position information. The sentence position information is embedded into the sentence with its centrality information together.

*POSES*: Extractive summarizer with the sentence position information. In this model, we use the sentence position information only to enhance sentence representation. The second block in the table shows the EDS models without the pre-trained encoder for comparison. The third block highlights the results of our model. The results show that the experimental performance on ROUGE-1, ROUGE-2, ROUGE-L outperforms all the models without pre-trained encoders.

**Table 2. The results of sentence centrality-enhanced EDS model based on HSG.**

| Model | ROUGE-1 | ROUGE-2 | ROUGE-L |
|---|---|---|---|
| LEAD-3 | 40.34 | 17.70 | 36.57 |
| JECS | 41.70 | 18.50 | 37.90 |
| LSTM+PN | 41.85 | 18.93 | 38.13 |
| HSG | 42.31 | 19.51 | 38.74 |
| HSG + Tri-Blocking | 42.95 | 19.76 | 39.23 |
| SCHSG | **43.01** | **19.98** | **39.39** |

The second block in the table shows the EDS models without the pre-trained encoder for comparison. The third block highlights the results of our model.

**Table 3. Performance difference caused by different sentence information.**

| Model | ROUGE-1 | ROUGE-2 | ROUGE-L |
|---|---|---|---|
| | CNN/Daily Mail | | |
| POSES | 43.23 | 20.23 | 39.60 |
| SCPES | 43.27 | 20.24 | 39.68 |
| SCES | **43.32** | **20.30** | **39.72** |
| | XSum | | |
| POSES | 23.67 | 4.60 | 17.89 |
| SCPES | 23.72 | 4.62 | 17.92 |
| SCES | **23.76** | **4.62** | **17.93** |

SCES is an extractive summarizer with the sentence centrality, SCPES is an extractive summarizer with the sentence and position information, POSES is an extractive summarizer with the sentence position information.

All the models in Table 3 are based on the pre-trained language model BERT, where the SCES model is exactly our SCBERT in Table 1. The various configurations of experimental parameters for the SCES model are the same as for the SCBERT model, except that the datasets used are different.

Table 3 shows the performance difference caused by the sentence position information and the sentence centrality. We can see that SCES performs well on news datasets CNN/Daily Mail and XSum. Combining the advantages of sentence centrality in reducing sentence-leading bias (we discuss it in the section Discussion) and experimental results, we can conclude that sentence centrality may be a better choice than sentence position information in the EDS task.

## Discussion

We argue that the effectiveness of sentence centrality is dataset-dependent. In news datasets, sentence position information can cause sentence-leading bias, which limits model performance. This problem is mitigated when sentence centrality replaces sentence position information.

We do a analysis driven by one question: why is sentence centrality a better choice than sentence position information in EDS tasks, especially on news datasets? According to the definition of the sentence centrality, sentences with higher centrality are more relevant to the document. Based on this, we calculated the top three sentence centrality scores distribution at different positions in the document.

Fig 4 shows that the distribution of top-3 sentence centrality scores in different positions. We can see that sentences with high centrality scores tend to be located in front of the document, especially the first three sentences, explaining why the Lead-3 model is so strong and effective. Sentence position information is a simplification of its centrality, because it cannot recognize the importance of sentences with high centrality scores but is located far from the first sentence. Another significant disadvantage of using positional information is that it is only valid on a particular dataset, such as news datasets.

Fig 5 shows the proportion of sentences extracted by different models in different positions in the test set. We used a greedy algorithm that is similar to Nallapati et al. [2] to obtain an ORACLE summary for each document. The algorithm generates an ORACLE consisting of multiple sentences by maximizing the ROUGE-2 score against the gold summary. For the sentences in the document, the ones in ORACLE will be marked with the label 1, and the others will be marked with the label 0. ORACLE summary is often used to train extractive models in extractive summarization task, because it represents the extraction upper bound. For

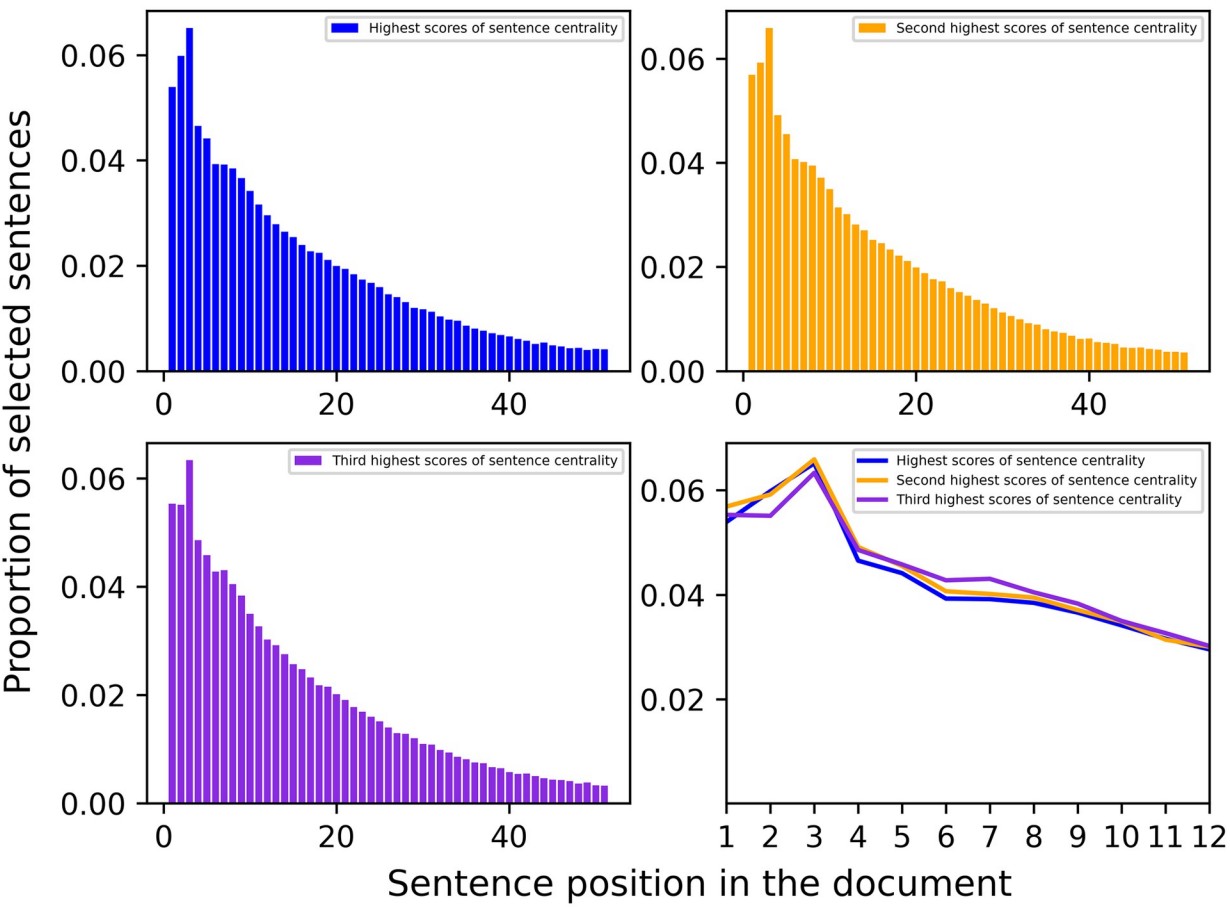

**Fig 4. The distribution of top-3 sentence centrality scores in different positions.** Sentences with high centrality scores tend to be located in front of the document.

comparison, we constructed the sentence centrality on the BERTSUM model with the sentence position information removed. Experimental results show that our model reduces the number of sentences in the front position and increases the number of sentences in the back position when forming summaries. A reason is that sentences at the front of the document but with lower centrality have a reduced impact on the model. Compared to models that use sentence position information, our model's outputs are more similar to ORACLE summaries.

## Conclusion

In this paper, we presented how sentence centrality can be usefully applied in two ways for improving extractive summarization performance. We introduced a novel way to calculate sentence centrality and proposed two approaches to applying sentence centrality to enhance sentence representation: (1) directly embedding sentence centrality into the sentence representation; (2) modifying the attention mechanism through sentence centrality. We revealed that the positional information of a sentence can be replaced by its centrality without introducing sentence-leading bias. In future work, we will continue to explore three points about sentence

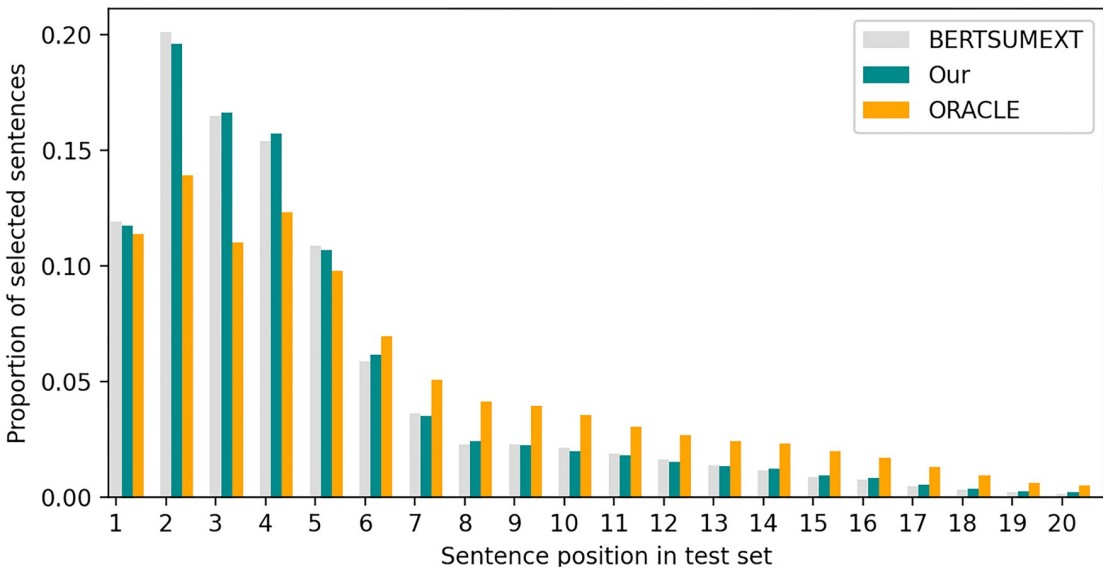

**Fig 5. Proportion of sentences extracted by different models in different positions.** BERTSUM is the BERT-based extractive summarization model with the sentence position. Oracle is the summary generated by the greedy algorithm.

centrality. First, the way we map scalar sentence centrality to a multi-dimensional space is straightforward. How to effectively model sentence centrality is worth exploring. Second, we will explore whether sentence centrality is also practical in other tasks, such as sentiment analysis, automatic question answering, etc. Finally, it would be useful to know how the proposed model performs with other similar node-local measures, such as the selectivity measure, which is also one of our future works.

## Supporting information

**S1 Appendix. Definition of ROUGE scores.**
(PDF)

## Acknowledgments

We would like to thank Professor Zhenfang Zhu for his guidance and support, who is also the corresponding author of the manuscript. We thank our NLP group for helpful discussion and valuable feedback on our paper. We also thank the reviewers for their patient and constructive review.

## Author Contributions

**Conceptualization:** Zhenfang Zhu, Jiangtao Qi, Chunling Tong.

**Data curation:** Chunling Tong, Qiang Lu, Wenqing Wu.

**Formal analysis:** Shuai Gong.

**Funding acquisition:** Zhenfang Zhu.

**Methodology:** Shuai Gong.

**Validation:** Jiangtao Qi.

**Writing – original draft:** Shuai Gong.

**Writing – review & editing:** Zhenfang Zhu.

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
