## [Decision Letter · Decision Letter 0]

23 Mar 2022

PONE-D-22-05139Improving Extractive Document Summarization with Sentence CentralityPLOS ONE

Dear Dr. Gong,

Thank you for submitting your manuscript to PLOS ONE. After careful consideration, we feel that it has merit but does not fully meet PLOS ONE’s publication criteria as it currently stands. Therefore, we invite you to submit a revised version of the manuscript that addresses the points raised during the review process.

We look forward to receiving your revised manuscript.

Kind regards,

Sanda Martinčić-Ipšić, PhD

Academic Editor

PLOS ONE

Journal Requirements:

Additional Editor Comments :

Please address reviewers comments adequately.

Reviewers' comments:

Reviewer's Responses to Questions

**Comments to the Author**

1. Is the manuscript technically sound, and do the data support the conclusions?

Reviewer #1: Yes

Reviewer #2: Partly

2. Has the statistical analysis been performed appropriately and rigorously? 

Reviewer #1: N/A

Reviewer #2: Yes

3. Have the authors made all data underlying the findings in their manuscript fully available?

Reviewer #1: Yes

Reviewer #2: Yes

4. Is the manuscript presented in an intelligible fashion and written in standard English?

Reviewer #1: Yes

Reviewer #2: Yes

5. Review Comments to the Author

Reviewer #1: The manuscript “Improving Extractive Document Summarization with Sentence Centrality” highlights a way to enhance sentence representation for extractive document summarization which in turns boost performance of existing EDS techniques. The paper is well written and contributions are clearly laid out and explained, used methodology is sound. Still some issues are cracking through. The following is a list of items which should be addressed:

1. In the introduction, the manuscript references findings of Zheng and Lapata that “similarity with the previous content will damage centrality”. In the method proposed here, forward edges are removed and only backward edges are considered, but this seems confusing. Why is this the right approach if this tends to damage centrality score?

2. Acronym EDS in line 10 is not defined – please add full text of the meaning.

3. The structure of the paper is missing at the end of Introduction.

4. While being sufficient example of a graph, figure 1 does not convey idea of degree centrality in a meaningful way. Maybe node size or colour should be varied based on centrality score to reinforce the idea that some sentences are more important than the other.

5. In the approach involving heterogeneous graph neural network, edge features were extended with sentence centrality. If sentence centrality is a node characteristic, why is it used to enhance edge features?

6. In the equation 6, indices in LHS appear to be duplicated. Equation 7 contains similar problem. Although it may be obvious to knowledgeable reader, terms of the equation 11 should still be defined for rigour, and it should be done for all other equations as well. For example, in equation (1) hi and hi+ are not defined.

7. It is not quite clear how is centrality embedding (EmbSCi) is obtained. Specifically, what are the terms on the RHS of the equation 9? If SCi is centrality of sentence i, what is the meaning of the exponents 1 to emb? Furthermore, equation 9 defines EmbSCi as a set of terms SCi, but this seems unusual w.r.t embeddings usually being vectors.

8. Please define HSG in line 168.

9. Please add short explanation of Trigram Blocking.

10. How does table 3 relate to table 1 and table 2? Are these various configurations of input data to SCBERT and SCHSG models from previous tables? If it is indeed so, please make sure it is more clear from the text to prevent any misunderstandings.

11. Please cite everything which is not original work presented in the paper, e.g. ROUGE, bert-base-uncased model, and some other instances.

12. Please in the appendix define used ROUGE scores.

13. “Analysis” section contains reference to “ORACLE summary”, please elaborate some more on what the ORACLE is, how does it work, and cite the relevant paper.

14. Consider renaming Analysis Chapter into Discussion and expand it.

15. Axes on figure 3 should be appropriately labelled to make the figure stands on it’s own.

16. It would be useful to know how the proposed model performs with other similar node-local measures such as selectivity measure. It might be useful as a basis for future work.

17. Introduction contains abbreviation “EDT” which is never elaborated or mentioned again. This seems like a very minor typographical error, and presumably was meant to say "EDS". In the same vein, “Ablation Study” section contains sentence “The results show that the experimental performance on ROUGE-L, ROUGE-2, ROUGE-L”. Is the first ROUGE-L in line 244 ROUGE-1?

18. On several places there is a blank after comma missing.

Reviewer #2: ### Overview and general recommendation

The paper addresses the problem of extractive document summarization. It uses sentence centrality information to enhance sentence representation. This information should reflect the sentence-document relationship and the sentence position information as well. The sentence representation is enhanced in two ways: one is embedded directly into the sentence representation output and the other updates the sentence representation indirectly via a graph attention network.

Because of advances in abstractive summarization, the task of extractive document summarization is probably no longer very challenging or interesting for the research field. The paper provides a good method for comparing the impact of sentence position vs. sentence centrality. However, one of the conclusions is not well supported by the experimental results, but overall, this is a very well-written paper with sound methodology, ablations studies, and a well-formulated research question.

### Major comments

- According to Table 3, the results of different sentence information don't support the conclusion that sentence centrality is a better choice than sentence position. Given that ROUGE is a poor metric, the difference of approximately 0.1 is insignificant.

- More ideas for future research could be added to the conclusion.

- The related research section could be expanded as well, e.g., adding a section on sentence embeddings, since the paper works on that level of representation.

- Some typos should be fixed, and some sentences could be rewritten to make them more clear; Take a look at minor remarks

- There is no link to the code repository.

### Minor comments

- Typo in the caption of Fig. 1, “documentt”

- Line 176: “using” → “use”

- Table 3: “summarizers” → “summarizer”

- Fig. 2: the first sentence should be corrected

- Line 244: the first “ROUGE-L” should be “ROUGE-1”

6. PLOS authors have the option to publish the peer review history of their article (what does this mean?). If published, this will include your full peer review and any attached files.

Reviewer #1: No

Reviewer #2: No

---

## [Author Response · Author response to Decision Letter 0]

18 Apr 2022

We greatly appreciate the reviewers taking the time to provide constructive comments and helpful suggestions. There is no doubt that the suggestions have significantly raised the quality if the manuscript and have enable us to improve the manuscript. Each suggested revision and comment brought forward by the reviewers was accurately incorporated and considered. We have carefully addressed all the reviewer's concerns. Please see our replies. Changes highlighted in red have been made accordingly in the revised manuscript.

Comments to the Author

1. Is the manuscript technically sound, and do the data support the conclusions? 

The manuscript must describe a technically sound piece of scientific research with 

data that supports the conclusions. Experiments must have been conducted rigorously, 

with appropriate controls, replication, and sample sizes. The conclusions must be 

drawn appropriately based on the data presented. 

Reviewer #1: Yes 

Reviewer #2: Partly

Response: Thanks for your review. We added experiments to verify the effectiveness of our method. We conducted experiments on XSum dataset. The results demonstrate the superiority of sentence centrality compared to positional information.

2. Has the statistical analysis been performed appropriately and rigorously? 

Reviewer #1: N/A 

Reviewer #2: Yes 

Response: Thank you for agreeing that our analysis is appropriate and rigorous.

3. Have the authors made all data underlying the findings in their manuscript fully 

available? 

The PLOS Data policy requires authors to make all data underlying the findings 

described in their manuscript fully available without restriction, with rare exception 

(please refer to the Data Availability Statement in the manuscript PDF file). The data 

should be provided as part of the manuscript or its supporting information, or 

deposited to a public repository. For example, in addition to summary statistics, the 

data points behind means, medians and variance measures should be available. If 

there are restrictions on publicly sharing data—e.g. participant privacy or use of data 

from a third party—those must be specified. 

Reviewer #1: Yes 

Reviewer #2: Yes 

Response: Thank you for your approval of our data.

4. Is the manuscript presented in an intelligible fashion and written in standard 

English? 

PLOS ONE does not copyedit accepted manuscripts, so the language in submitted 

articles must be clear, correct, and unambiguous. Any typographical or grammatical 

errors should be corrected at revision, so please note any specific errors here. 

Reviewer #1: Yes 

Reviewer #2: Yes

Response: Thank you for your approval.

Review Comments to the Author

Please use the space provided to explain your answers to the questions above. You 

may also include additional comments for the author, including concerns about dual 

publication, research ethics, or publication ethics. (Please upload your review as an 

attachment if it exceeds 20,000 characters) 

Reviewer #1: The manuscript “Improving Extractive Document Summarization with 

Sentence Centrality” highlights a way to enhance sentence representation for 

extractive document summarization which in turns boost performance of existing EDS techniques. The paper is well written and contributions are clearly laid out and 

explained, used methodology is sound. Still some issues are cracking through. The 

following is a list of items which should be addressed: 

1. In the introduction, the manuscript references findings of Zheng and Lapata that 

“similarity with the previous content will damage centrality”. In the method proposed here, forward edges are removed and only backward edges are considered, but this seems confusing. Why is this the right approach if this tends to damage centrality score?

Response: We appreciate the reviewer for asking questions about the details of sentence centrality. According to the original paper of Zheng and Lapata, the centrality score of s_i based on the directed graph can be defined as follows:

centrality(s_i )=λ_1 ∑_(j<i)e_ij +λ_2 ∑_(j>i)e_ij ,

where the optimal λ_1 tends to be negative.

In our paper, we do not calculate the similarity between the sentence and its previous content of the sentence, which means that we set the weights of forward-looking directed edges λ_1 are equal to 0. 

We think the descriptions of “forward-looking” and “forward” make our point unclear, so we revised this part of content to convey our idea clearly, in lines 43-45. 

The description of “Inspired by their work, we remove the forward edges of sentences on directed graphs and calculate the sentence centrality based only on the weights of backward edges” is modified by us to “Inspired by their work, we calculate the centrality score of a sentence based only on the similarity between the sentence and its following content”.

2. Acronym EDS in line 10 is not defined – please add full text of the meaning.

Response: We appreciate the reviewer for his/her careful review. We have added the full text of the meaning of EDS in line 10.

3. The structure of the paper is missing at the end of Introduction.

Response: We thank the reviewer for reminding us to describe the structure of our paper, and there is no doubt that this suggestion makes the paper more readable. We have added a description of the paper structure in lines 76-81.

4. While being sufficient example of a graph, figure 1 does not convey idea of degree 

centrality in a meaningful way. Maybe node size or colour should be varied based on 

centrality score to reinforce the idea that some sentences are more important than the other.

Response: We thank the reviewer for the helpful suggestions on our figure 1. We now updated the figure 1. we used different colors to indicate different sentences, and used the size of the nodes to indicate the centrality scores. We also increased the number of sentence nodes in order to convey idea our ideas more clearly. 

5. In the approach involving heterogeneous graph neural network, edge features were 

extended with sentence centrality. If sentence centrality is a node characteristic, why 

is it used to enhance edge features?

Response: We are grateful to the reviewer for questions about our method. 

We use sentence centrality to enhance edge features is to modify the graph attention (GAT) layer. In the heterogeneous graph neural network, the sentence representations are updated with their neighbor word nodes via a GAT layer and feed-forward (FFN) layer. The GAT layer is modified by infusing the scalar edge weights e_ij (described in equation 13), which are mapped to the multi-dimensional embedding space. The weights of the edge e_ij are the sum of the sentence centrality and the TF-IDF value of the words, because the types of nodes connected by the edge are different.

We added equations and a textual explanation to describe how sentence representations are updated by GAT and FFN in equation 17, equation 18 and lines 228-236. We also do a textual explanation of why we use sentence centrality to enhance edge features in lines 216-221.

6. In the equation 6, indices in LHS appear to be duplicated. Equation 7 contains 

similar problem. Although it may be obvious to knowledgeable reader, terms of the 

equation 11 should still be defined for rigour, and it should be done for all other 

equations as well. For example, in equation (1) hi and hi+ are not defined.

Response: We appreciate the reviewer for his/her careful review and we feel sorry for our lack of rigour. 

We have corrected equation 6 and equation 7. For terms in the equations that are not strictly defined, we have carefully checked.

For equation 1, we defined h_i and h_i^+, and explained the meaning of sim(h_i,h_i^+) in lines 150-152.

For equation 6, we explained the meaning of the term u_ij in line 184.

For equation 9 and equation 10, we modified these two equations to make the meaning of 〖EmbSC〗_i clearer. We defined each term in lines 187-192.

For equation 11, we define each term of the equation and explain the meaning in lines198-200. 

7. It is not quite clear how is centrality embedding (EmbSCi) is obtained. Specifically, 

what are the terms on the RHS of the equation 9? If SCi is centrality of sentence i, 

what is the meaning of the exponents 1 to emb? Furthermore, equation 9 defines 

EmbSCi as a set of terms SCi, but this seems unusual w.r.t embeddings usually being 

vectors.

Response: We appreciate the reviewer for his/her careful review. Our definition of equation 9 is not rigorous, leaving our point unclear. We feel sorry for this. We modified the equation 9. Now the equation 9 is:

EmbSC_i=W_sc (SC_i ) ,

where W_sc is a weight matrix with the weights set to 1. EmbSC_i is the centrality embedding of sentence s_i, which has the same dimension as the sentence embedding.

EmbSC_i is obtained by mapping the normalized scalar sentence centrality to the multi-dimensional embedding space. The RHS of the new equation 9 W_sc (SC_i ) means that (SC_i ) is mapped to a higher dimensional space by W_sc. The RHS of our previous equation 9 was trying to convey the same meaning as the new equation 9, but we are sorry that we did not make it clearer. The exponents 1 to emb means that we map sentence centrality to the emb-dimensional space in the previous equation 9. 

8. Please define HSG in line 168.

Response: We thank the reviewer for reminding us to define HSG. We add full text of the meaning HSG in lines 202-204, and give our sentence centrality-enhanced extractive document summarization model based on HSG in Fig.3.

9. Please add short explanation of Trigram Blocking.

Response: We thank the reviewer for reminding us to add short explanation of Trigram Blocking. We added the short explanation of Trigram Blocking in lines 270-273. 

10. How does table 3 relate to table 1 and table 2? Are these various configurations of 

input data to SCBERT and SCHSG models from previous tables? If it is indeed so, 

please make sure it is clearer from the text to prevent any misunderstandings.

Response: We thank the reviewer for his/her careful review. All the models in Table 3 are based on the pre-trained language model BERT, where the SCES model is exactly our SCBERT in Table 1. The various configurations of experimental parameters for the SCES model are the same as for the SCBERT model, except that the datasets used are different. We added the relevant description in lines 324-327.

11. Please cite everything which is not original work presented in the paper, e.g. 

ROUGE, bert-base-uncased model, and some other instances.

Response: We thank the reviewer for his/her careful review. We checked our paper carefully and added citations to the work that needed to be cited.

Line 17, we added a reference to transformer.

Line 213, we added a reference to Convolutional Neural Network. 

Line 216, we added a reference to Bidirectional Long and Short-Term Memory.

Line 239, we added references to CNN/Daily Mail and XSum datasets.

Line 297, we added references to ROUGE.

“bert-base-uncased model” is published in https://github.com/huggingface/pytorch-pretrained-BERT, we put the URL in lines 256-257.

12. Please in the appendix define used ROUGE scores.

Response: We thank the reviewer for reminding us to add the definition of ROUGE scores. We defined the used ROUGE scores in S1 Appendix.

13. “Analysis” section contains reference to “ORACLE summary”, please elaborate 

some more on what the ORACLE is, how does it work, and cite the relevant paper.

Response: We thank the reviewer for his/her rigorous review. We elaborated the ORACLE summary in lines 353-359, including that what the ORACLE is, how does it work. We also cite the relevant paper.s

14. Consider renaming Analysis Chapter into Discussion and expand it.

Response: We thank the reviewers for the suggestions on the structure of our article.

We have renamed Analysis Chapter into Discussion. In this part, we added the content of ORACLE according to your comment 13. We discuss why sentence centrality is a better choice than sentence position information.

15. Axes on figure 3 should be appropriately labelled to make the figure stands on it’s 

own.

Response: We thank the reviewer for his/her careful review. Since we added a heterogeneous graph model graph, the original figure 3 is now figure 4. The axes on figure 4 are now appropriately labeled.

16. It would be useful to know how the proposed model performs with other similar 

node-local measures such as selectivity measure. It might be useful as a basis for 

future work.

Response: We thank the reviewer for his/her constructive suggestion. This suggestion makes us realize that our exploration of sentence centrality needs to go further. We have written this suggestion into future work. Thanks again for the constructive suggestion.

17. Introduction contains abbreviation “EDT” which is never elaborated or mentioned 

again. This seems like a very minor typographical error, and presumably was meant to 

say "EDS". In the same vein, “Ablation Study” section contains sentence “The results 

show that the experimental performance on ROUGE-L, ROUGE-2, ROUGE-L”. Is 

the first ROUGE-L in line 244 ROUGE-1?

Response: We appreciate the reviewer for his/her careful review. “EDT” is a typographical error; we have corrected it in line 41. The first “ROUGE-L” is corrected to “ROUGE-1” in line 306.

18. On several places there is a blank after comma missing.

Response: We appreciate the reviewer for his/her careful review. We checked our paper carefully and added a blank for commas.

We again thank the reviewer for taking the time to review our article. From the review comments, we can feel the rigorous attitude of the reviewers to academics. There is no doubt that the comments of the reviewers make our manuscript more rigorous and clearer. 

Reviewer #2: ### Overview and general recommendation 

The paper addresses the problem of extractive document summarization. It uses 

sentence centrality information to enhance sentence representation. This information 

should reflect the sentence-document relationship and the sentence position 

information as well. The sentence representation is enhanced in two ways: one is 

embedded directly into the sentence representation output and the other updates the 

sentence representation indirectly via a graph attention network. 

Because of advances in abstractive summarization, the task of extractive document 

summarization is probably no longer very challenging or interesting for the research 

field. The paper provides a good method for comparing the impact of sentence 

position vs. sentence centrality. However, one of the conclusions is not well supported 

by the experimental results, but overall, this is a very well-written paper with sound 

methodology, ablations studies, and a well-formulated research question. 

### Major comments 

- According to Table 3, the results of different sentence information don't support the 

conclusion that sentence centrality is a better choice than sentence position. Given 

that ROUGE is a poor metric, the difference of approximately 0.1 is insignificant.

Response: We thank the reviewer for pointing out potential limitation in our study. 

We agree with the reviewer that the superiority of sentence centrality cannot be demonstrated only from the ROUGE scores.

In the extractive summarization task, position information is usually used to enhance sentence representation. Although doing so will improve model extraction performance significantly, it will cause sentence-leading bias, especially in news datasets. We present this phenomenon in lines 23-30.

we replaced the sentence position information with sentence centrality to reduce sentence-leading bias without causing model performance degradation, which can be seen in the figure 5 and table 3. We added experiments in the news dataset XSum. The experimental results show that the replacement of sentence position information by sentence centrality will not cause model performance degradation.

Before reaching the conclusion "sentence centrality is a better choice than sentence position", we added a description of the advantages of sentence centrality in reducing sentence leading bias, discussed in the Discussion section.

Combining the advantages of sentence centrality in reducing sentence leading bias and experimental results in table 3, we can conclude that sentence centrality information has certain advantages over sentence position information the extractive summarization task.

We are grateful to the reviewer for his/her constructive comments, which made our logic more rigorous and greatly improved the quality of our paper. 

- More ideas for future research could be added to the conclusion.

Response: We thank the reviewer for reminding us to expand our future work. We add more ideas in future research, including exploring whether sentence centrality is also effective in other tasks, etc., which are presented in lines 374-381. 

- The related research section could be expanded as well, e.g., adding a section on 

sentence embeddings, since the paper works on that level of representation. 

Response: We thank the reviewer for his/her constructive suggestion. Adding a section on sentence embeddings will make our article more rigorous. We have expanded our related work in lines 113-131.

- Some typos should be fixed, and some sentences could be rewritten to make them. Take a look at minor remarks

Response: We thank the reviewer for his/her careful review. We have carefully checked our article and corrected errors. We present the modification details in the ### Minor comments section.

- There is no link to the code repository.

Response: We are pleased that the reviewer is interested in our work.

The code is released at https://github.com/GongShuai8210/SCES.

### Minor comments 

- Typo in the caption of Fig. 1, “documentt” 

- Line 176: “using” → “use” 

- Table 3: “summarizers” → “summarizer” 

- Fig. 2: the first sentence should be corrected 

- Line 244: the first “ROUGE-L” should be “ROUGE-1”

Response: We thank the reviewer for taking the time to review our article carefully. We have corrected the typo now.

-Typo in the caption of Fig. 1, “documentt” → “document”

- Line 213: “using” has been corrected to “use” 

- Table 3: “summarizers” has been corrected to “summarizer” 

- Fig. 2: the first sentence has been corrected to “EmbSC_i is the centrality embedding of sentence s_i, which is directly embedded in the sentence representation generated by BERT”. 

- Line 306: the first “ROUGE-L” has been corrected to “ROUGE-1”.

We again thank the reviewer for his/her careful review and constructive suggestions. There is no doubt that the reviewer's suggestions improve the quality of our article.

---

## [Editor Report · Decision Letter 1]

27 Apr 2022

Improving Extractive Document Summarization with Sentence Centrality

PONE-D-22-05139R1

Dear Dr. Gong,

We’re pleased to inform you that your manuscript has been judged scientifically suitable for publication and will be formally accepted for publication once it meets all outstanding technical requirements.

Kind regards,

Sanda Martinčić-Ipšić, PhD

Academic Editor

PLOS ONE

Additional Editor Comments (optional):

I have reviewed the revised manuscript, responses to reviewer comments, and availability of data and SW. The authors have addressed all reviewer comments and improved the quality of the manuscript. I am pleased to report that the current manuscript revision adequately addresses all issues and meets the required PlosOnNE criteria.
---

## [Editor Report · Acceptance letter]

14 Jul 2022

PONE-D-22-05139R1 

Improving Extractive Document Summarization with Sentence Centrality 

Dear Dr. Zhu:

I'm pleased to inform you that your manuscript has been deemed suitable for publication in PLOS ONE. Congratulations! Your manuscript is now with our production department. 

Kind regards, 

on behalf of

Dr. Sanda Martinčić-Ipšić 

Academic Editor

PLOS ONE